# EvFocus: Learning to Reconstruct Sharp Images from Out-of-Focus Event Streams

**Lin Zhu**[1] **Xiantao Ma**[1] **Xiao Wang**[2] **Lizhi Wang**[3] **Hua Huang**[3]

## Abstract

Event cameras are innovative sensors that capture brightness changes as asynchronous events rather than traditional intensity frames. These cameras offer substantial advantages over conventional cameras, including high temporal resolution, high dynamic range, and the elimination of motion blur. However, defocus blur, a common image quality degradation resulting from out-of-focus lenses, complicates the challenge of event-based imaging. Due to the unique imaging mechanism of event cameras, existing focusing algorithms struggle to operate efficiently on sparse event data. In this work, we propose EvFocus, a novel architecture designed to reconstruct sharp images from defocus event streams for the first time. Our work includes the development of an event-based out-of-focus camera model and a simulator to generate realistic defocus event streams for robust training and testing. EvFocus integrates a temporal information encoder, a blur-aware two-branch decoder, and a reconstruction and re-defocus module to effectively learn and correct defocus blur. Extensive experiments on both simulated and real-world datasets demonstrate that EvFocus outperforms existing methods across varying lighting conditions and blur sizes, proving its robustness and practical applicability in event-based defocus imaging.

## 1. Introduction

Event cameras are innovative sensors that capture brightness changes in the form of asynchronous events rather than traditional intensity frames. These cameras offer significant

[1]School of Computer Science& Technology, Beijing Institute of Technology, Beijing, China [2]School of Computer Science, Anhui University, Hefei, China [3]School of Artificial Intelligence, Beijing Normal University, Beijing, China. Correspondence to: Hua Huang <huahuang@bnu.edu.cn>.

*Proceedings of the 42nd International Conference on Machine Learning*, Vancouver, Canada. PMLR 267, 2025. Copyright 2025 by the author(s).

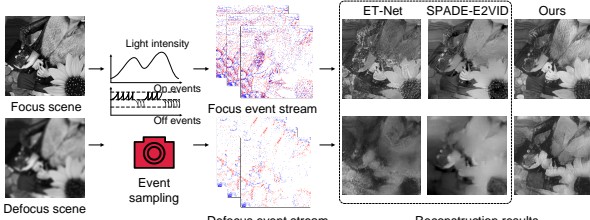

*Figure 1.* Reconstructing sharp images from defocus event streams. Our model reconstructs sharp images that closely approximate the in-focus results from defocus event streams, effectively mitigating the impact of defocus blur in event-based imaging.

advantages, such as high temporal resolution, high dynamic range, and immunity to motion blur, making them particularly useful in scenarios where conventional cameras struggle (Gallego et al., 2020b). The ability to capture precise and rapid changes in brightness opens new possibilities for image reconstruction applications, particularly in challenging conditions like low light and high-speed motion (Delbruck & Lichtsteiner, 2007), where traditional cameras often fail to deliver clear images.

Recent research increasingly leverages event data to enhance image and video deblurring, capitalizing on the rich edge information provided by event streams. Methods like (Jiang et al., 2020), (Lin et al., 2020), and (Wang et al., 2020) have successfully integrated event data with RGB image and video deblurring, leading to notable improvements in performance. For instance, (Sun et al., 2022) employs a multi-head attention mechanism to combine event and image data, while (Kim et al., 2022) introduces an exposure time-based event selection (ETES) module to handle images with unknown exposure times. Additionally, (Lin et al., 2022) proposes an autofocus strategy utilizing event streams, though it struggles with fixed focus scenarios. Meanwhile, (Teng et al., 2024) develops a method using event accumulation during continuous focus scanning to aid RGB cameras in predicting sharp, focused images.

However, a major limitation of existing methods is their inability to effectively address defocus blur, which is a common and critical issue in many real-world scenarios (see Fig. 1). Unlike traditional cameras, which benefit from mature autofocus technology, event cameras lack such mecha-

nisms and often rely on manual focusing, making defocus a frequent occurrence in event data. While motion blur is naturally mitigated by the asynchronous nature of event cameras, defocus blur fundamentally reduces spatial intensity gradients, resulting in a significant decrease in the number and quality of generated events. This makes defocus blur particularly challenging to handle within existing event-based frameworks, as the temporal dynamics of event streams are directly affected by the loss of gradient information. Unlike traditional deblurring techniques for RGB images, where spatial restoration dominates, the sparsity and continuous nature of event streams necessitate a temporal-domain approach to effectively handle defocus blur.

In this paper, we present EvFocus, a novel framework specifically designed to reconstruct sharp images from defocus event streams. To support this, we first develop a theoretical model of defocus blur in event cameras and a simulator capable of generating large-scale, realistic defocus event datasets. These datasets include event streams, blurred images, sharp images, and optical flow, covering a wide range of defocus conditions. Unlike traditional defocus blur datasets, which primarily focus on spatial restoration, our dataset and approach emphasize the temporal characteristics of defocus event streams, bridging the gap between theoretical exploration and practical application. Our framework includes a temporal information encoder with ConvLSTM layers to capture and preserve temporal dependencies, and a blur-aware two-branch decoder that separates blur-specific and alignment features to ensure precise reconstruction. Additionally, we design a reconstruction and re-defocus module, where the aligned feature passes through a Re-defocus module and a Reconstruction module. The Re-defocus Image generated from this process undergoes self-supervised learning with the blur feature, allowing the model to further learn defocus blur distribution. Experimental results demonstrate that our architecture effectively eliminates defocus blur across different conditions. In summary, the contributions of this work are:

1) We propose EvFocus, the first network designed to effectively reconstruct sharp images from defocus event streams, featuring a temporal information encoder, a blur-aware two-branch decoder, and a reconstruction and re-defocus module to ensure precise image reconstruction.

2) We explore and model defocus events in event cameras and develop a simulator to generate realistic defocus event streams for robust training and evaluation. Additionally, we introduce a defocus feature learning mechanism that enhances the model's ability to learn defocus blur distribution through the blur-aware two-branch decoder.

3) Extensive experiments on both simulated and real-world defocus event data demonstrate the superior performance of EvFocus, outperforming existing methods across various

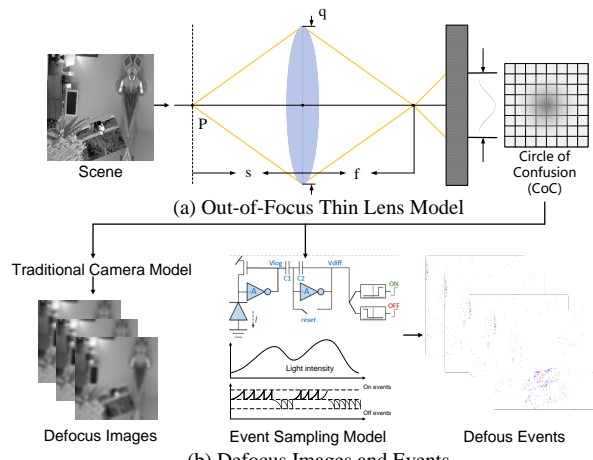

Figure 2. The sampling mechanism of defocus events. When the lens is out-of-focus, light is focused outside the sensor through the thin lens model and forms a CoC on the sensor. When the light change at a single pixel on the sensor reaches the threshold, the defocus event is generated.

lighting conditions and blur sizes, proving its robustness.

## 2. Related Work

**Traditional Image-based Defocus Deblurring.** Defocus deblurring in RGB images follows two main approaches: defocus map estimation with deconvolution and deep learning-based methods. Early works (Yi & Eramian, 2016; D'Andrès et al., 2016) estimate defocus maps before applying deconvolution, while (Xin et al., 2021) introduces an unsupervised approach with calibrated blur kernels. Deep learning has significantly improved defocus deblurring, with methods like (Abuolaim & Brown, 2020; Lee et al., 2021; Son et al., 2021; Cho et al., 2021a; Ruan et al., 2022) enhancing accuracy and efficiency.

For video deblurring, integrating temporal information helps recover sharp details from multiple frames (Chen et al., 2018; Pan et al., 2023; Zhu et al., 2022a). However, these methods rely on intensity-based images and do not address event-based defocus blur. Unlike existing defocus blur datasets focused on spatial restoration, we explore the temporal characteristics of defocus-induced event streams, shifting the focus from frame-based deblurring to event-based reconstruction.

**Event-based Reconstruction.** Neuromorphic cameras offer low latency and energy efficiency (Delbruck & Lichtsteiner, 2007; Zhu et al., 2019; 2020). Event-based reconstruction, which recovers intensity images or video from asynchronous events, is a core yet challenging application. Event-based reconstruction has evolved from early physics-driven methods to deep learning. Classical approaches (Cook et al., 2011; Kim et al., 2008; Bardow et al., 2016) rely on optimization

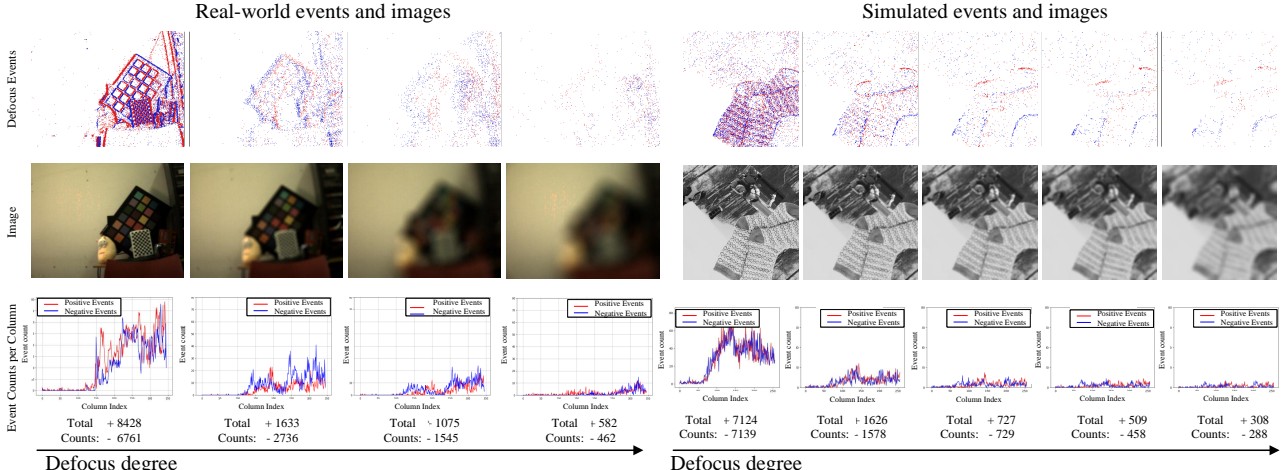

*Figure 3.* Event distribution of real-world (Left) and simulated (Right) data with different degrees of blur. We present the visual effects of RGB images under different degrees of blur as well as the corresponding events and event distribution statistics.

and filtering to estimate intensity images, while (Munda et al., 2018; Barua et al., 2016) shows motion-independent reconstruction feasibility. Deep learning further improves event-based reconstruction. E2VID (Rebecq et al., 2019) introduces a recurrent network using voxel grids, while FireNet (Scheerlinck et al., 2020) achieves faster inference but struggles with high-speed motion. Improvements like E2VID+ and FireNet+(Stoffregen et al., 2020) align synthetic and real data, and SPADE-based(Cadena et al., 2021) and transformer-based (Weng et al., 2021) architectures refine quality. Recent advances, including SNN-based (Zhu et al., 2022b) and dynamic event reconstruction models (Ercan et al., 2024), enhance efficiency and adaptability.

**Event-based Deblurring.** Events have been leveraged to enhance deblurring by preserving high-temporal-resolution edges (Jiang et al., 2020; Lin et al., 2020; Wang et al., 2020). Methods like (Sun et al., 2022) use multi-head attention to fuse event and image features, while (Kim et al., 2022) proposes an exposure time-based selection module. Autofocus strategies (Lin et al., 2022) remain limited to fixed-focus cases, and (Teng et al., 2024) introduces event accumulation for continuous focus scanning. (Lou et al., 2023) employ the event focal stack method to assist RGB image refocusing; however, this approach imposes stringent constraints on data acquisition.

Despite progress, event-based defocus deblurring remains challenging. Unlike motion blur, which events naturally mitigate, defocus blur weakens spatial gradients, reducing event generation and complicating sharp image reconstruction. Factors like blur kernel size, event firing threshold, and refractory period further hinder effective modeling. Addressing this requires a temporal-domain approach to capture defocus event dynamics, moving beyond conventional spatial deblurring techniques.

## 3. Background and Motivation

**Event Sampling Mechanism.** The sensor of the event camera operates independently at each pixel to detect changes in light intensity. An event is generated when the light intensity at pixel changes by more than a certain threshold from the time the last event is recorded (Gallego et al., 2020a). This can be expressed as:

$$\Delta L = \log I(t) - \log I(t - \Delta t), \qquad (1)$$

where an event is generated if $|\Delta L| > C$, $C$ denotes the threshold. An event is defined by a quadruplet which contains the event's timestamp $t$, the pixel coordinates $(x, y)$, and the event polarity $p$.

**Event-based Out-of-Focus Camera Model.** Suppose that the $k$-th event in an event stream is expressed as

$$E_k = (x_k, y_k, t_k, p_k). \qquad (2)$$

At $t_k$, changes in light intensity at the pixel reaches the threshold $C$, Eq. 1 can be rewritten as

$$\Delta L(x_k, y_k, t_k) = L(x_k, y_k, t_k) - L(x_k, y_k, t_k - \Delta t_k), \qquad (3)$$

where $|\Delta L(x_k, y_k, t_k)| \geq C$. Conventional images typically employ the thin lens model to model defocus blur (Mannan & Langer, 2016). This model presumes that the thickness of the lens can be ignored, which is beneficial for simplifying optical ray tracing computations. Through this model, we can approximate the circle of confusion (CoC) of the PSF of a given point (Levin et al., 2007) based on the distance of the point from the lens and the camera parameters (i.e., focal length, aperture size, and distance). The model is shown in Fig. 2, where $f$ is the focal length, $s$ is the object distance, and $F$ is the aperture value. The

distance between the lens and the sensor $s'$ and the aperture diameter $q$ are defined as: $s' = \frac{fs}{s-f}, \quad q = \frac{f}{F}$.

The CoC radius $r$ of the scene point $P_1$ at distance $d$ from the camera is given by:

$$r = \frac{q}{2} \times \frac{s'}{s} \times \frac{d-s}{d}. \quad (4)$$

Thus, defocused RGB imaging can be modeled as:

$$I_{\text{blur}}(x, y) = \big(I * h\big)_{(x,y)},$$

Here, $I * h$ represents the convolution operation, where $I$ is the input image and $h$ is the convolution kernel.

Considering the diffraction of light within the CoC of defocus blur, the light intensity can be approximated as a Gaussian distribution (Gokstorp, 1994; Elder & Zucker, 1998). This results in blurred edges and degraded image quality in RGB camera captures. To model event-based defocus blur, we approximate it using Gaussian blur. The defocused temporal light intensity difference $\Delta L'$ is defined as:

$$\Delta L'(x_k, y_k, t_k) = L'(x_k, y_k, t_k) - L'(x_k, y_k, t_k - \Delta t_k), \quad (5)$$

where $L'(x_k, y_k, t_k)$ and $L'(x_k, y_k, t_k - \Delta t)$ represents $\big(L * G_{t_k}\big)_{(x_k, y_k, t_k)}$ and $\big(L * G_{t_k - \Delta t}\big)_{(x_k, y_k, t_k - \Delta t)}$, respectively. Here, $G_{t_k}$ represents the Gaussian kernel at time $t_k$, which approximates the size of the CoC. The blur kernel at $(x_k, y_k)$ and timestamp $t_k$ can be modeled by a Gaussian distribution and is expressed as $G(x, y) = \frac{1}{2\pi\sigma^2} \exp\left(-\frac{x^2+y^2}{2\sigma^2}\right)$.

**Distribution of Out-of-Focus Event Streams.** As the object distance and focal length are continuous variables, and the time resolution of the event camera is extremely high, we can assume that $G$ remains unchanged at this moment, i.e., $G_{t_k}(u, v) = G_{t_k - \Delta t}(u, v)$. Therefore,

$$\Delta L'(x_k, y_k, t_k) = \sum_{u=-s}^{s} \sum_{v=-s}^{s} \Delta L(u, v) \cdot G(u, v), \quad (6)$$

where $\Delta L(u, v) = L(x+u, y+v, t_k) - L(x+u, y+v, t_k - \Delta t)$ and $s$ represents the size of the CoC at point $(x_k, y_k)$. When a pixel lies at a local strong edge or experiences a significant local brightness change, events are typically triggered. However, the defocus PSF causes the brightness peaks to shrink and the valleys to rise, resulting in local smoothing (defocus averaging). Due to the presence of this blurring phenomenon, strong brightness changes degrade into weaker brightness changes, which statistically manifests as a reduction in event triggering and a degradation of the event distribution into a blurred distribution.

For instance, consider a $3 \times 3$ CoC where only the central pixel experiences a brightness change that exceeds the event threshold, while all other pixels do not trigger events. In this case, under normal focus, the brightness change at the central pixel $(x_k, y_k)$ satisfies $\Delta L(x_k, y_k, t_k) > C$.

After defocus, the brightness change is averaged across the CoC, as described in Eq. 6. Since only the central pixel contributes a significant brightness change, the defocus effect reduces the overall brightness change at $(x_k, y_k)$, and we have $\Delta L'(x_k, y_k, t_k) < C$. As a result, no events are triggered due to the "Defocus Averaging", even though the original brightness change exceeded the threshold. Thus, we get $\Delta L'(x_k, y_k, t_k) \leq \Delta L(x_k, y_k, t_k)$.

Equality is achieved when all pixels within the blur kernel exhibit brightness changes with the same polarity, resulting in no cancellation effects. More generally, when pixels within the blur kernel exhibit brightness changes of mixed polarities, the contributions of positive and negative changes are weighted by the blur kernel $G$. For instance, if $x_k$ emits a positive event and another point $x_r$ within the kernel emits a negative event with $|\Delta L_{x_r}| < |\Delta L_{x_k}|$, then $\Delta L_{x_r} < 0$ and the weighted contribution $\Delta L'_{x_r} > 0$ remains within the overall summation.

From the above, we know that defocus blur causes events to become sparse (Fig. 3). The smaller the area of the object generating events, the more significant the effect of defocus blur on event generation. When events with both positive and negative polarities are generated within the blur kernel area, the polarity with smaller brightness changes is more affected by defocus blur, resulting in either no event being generated or an event generated with the opposite polarity. For further analysis, please refer to the supplementary material.

**Out-of-Focus Event Stream Simulation.** We develop a simulator to generate realistic defocus event streams for robust training and evaluation. In supplementary material, Alg. 1 presents the simulation of our out-of-focus event stream. We apply Gaussian blur to the simulated event data using the Multi-Objects-2D renderer option of ESIM (Rebecq et al., 2018) where multiple moving objects are captured with a camera restricted to 2D motion. To simulate varying levels of defocus in real-world scenarios, we employ different PSF configurations for each scene. Experimental results demonstrate that the model trained on synthetic data achieves effective reconstruction for various defocus levels observed in real-world scenarios. We use the configuration of (Stoffregen et al., 2020) to ensure that our data has a similar distribution as real-world data. We approximate defocus blur with multi-levels gaussian filter which is applied when high temporal resolution frames are rendered.

## 4. The Proposed Model

Based on previous analysis, unlike traditional image defocus, event defocus also alters the number of events, mak-

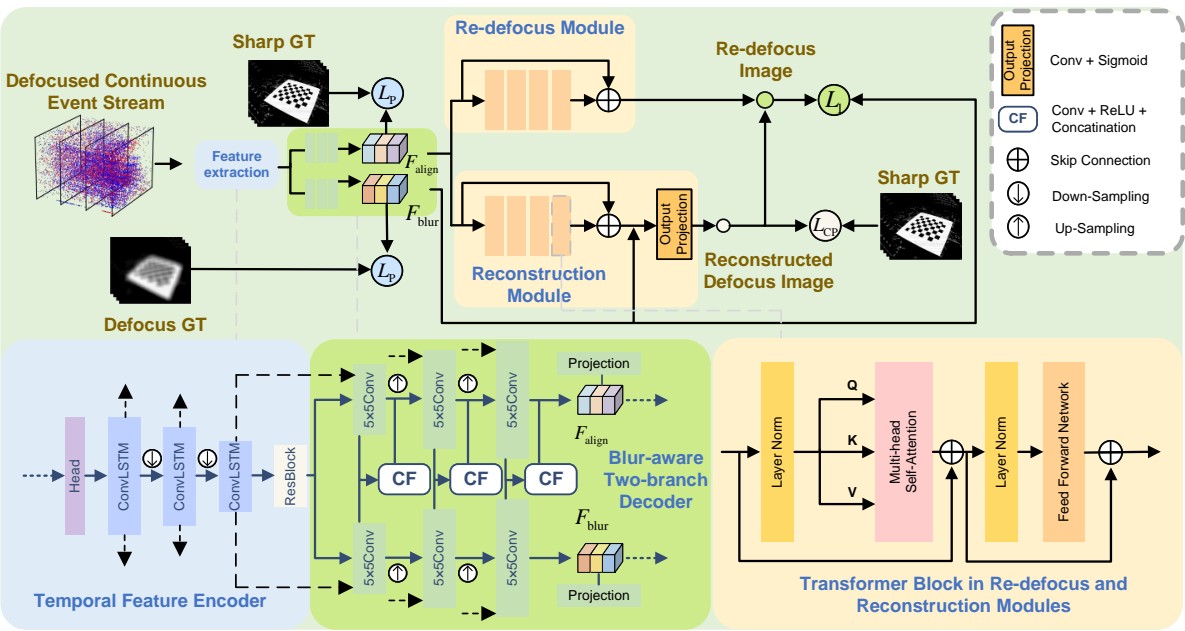

*Figure 4.* The architecture of EvFocus, includes a Temporal Information Encoder, a Blur-aware Two-branch Decoder for separating blur and alignment features, and a Reconstruction and Re-Defocus Module that refines defocus blur correction through self-supervised learning, resulting in precise image reconstruction from defocus event streams.

ing event defocus deblurring and reconstruction a highly ill-posed problem. In this section, we propose EvFocus, specifically designed to address the unique challenges posed by event cameras for deblurring and reconstructing defocus event streams (see Fig. 4). The model first extracts the temporal feature from the defocus event stream, then introduces a two-branch decoder and reconstruction and re-defocus module, enabling robust defocus blur correction and image reconstruction. The model is structured as follows:

**Temporal Information Encoder.** The event camera's asynchronous nature produces sparse event streams rather than conventional frame-based sequences, making it difficult to extract meaningful temporal information. Our temporal information encoder is motivated by the need to capture and retain temporal dependencies in defocus event streams without losing critical temporal resolution. Similar to E2VID (Rebecq et al., 2019), we utilize ConvLSTM units to effectively preserve temporal information over time by maintaining hidden states.

**1) Event Representation.** Events are typically processed and converted into alternative representations to fit the network. In our model, we utilize voxel grid (Zihao Zhu et al., 2018) as the spatio-temporal representation of event stream.

**2) Feature Extraction.** To stabilize the feature extraction process in an asynchronous event data context, we incorporate ConvLSTM layers. These layers are chosen because

they can selectively forget irrelevant features and remember important temporal information over multiple time steps, enhancing the network's ability to learn and extract useful spatio-temporal features from the sparse event data.

Each encoder layer consists of a 2D down-sampling convolution and a ConvLSTM (Shi et al., 2015). The 2D down-sampling convolution has a kernel size of 5 and a stride of 2. The ConvLSTM has a kernel size of 3, and the number of input and hidden layers is the same as the down-sampling convolution. Each encoder maintains a state $s_i^k$ that is updated at each iteration to memorize temporal information. ConvLSTM can enhance the stability of features by remembering and forgetting previous states. Three consecutive ConvLSTM blocks can be represented by the equation :

$$\mathbf{f}_i^{CL}, \mathbf{s}_i^t = f_l^{rec}(\mathbf{f}_{i-1}^{CL}, \mathbf{s}_i^{t-1}), \qquad (7)$$

where $\mathbf{f}_i^{CL}$ denote the feature of ConvLSTM, $i \in \{1, 2, 3\}$ denotes the current $i$-th ConvLSTM block, and $\mathbf{s}_i^t$ denotes the state of the $i$-th ConvLSTM block at time $t$.

**Blur-aware Two-branch Decoder.** The blur-aware two-branch decoder addresses the challenge of separating blur-specific features from the alignment features necessary for accurate image reconstruction. The need for two parallel decoding branches arises from the observation that blur and alignment features often interfere with each other during reconstruction. By processing them separately, we can ensure

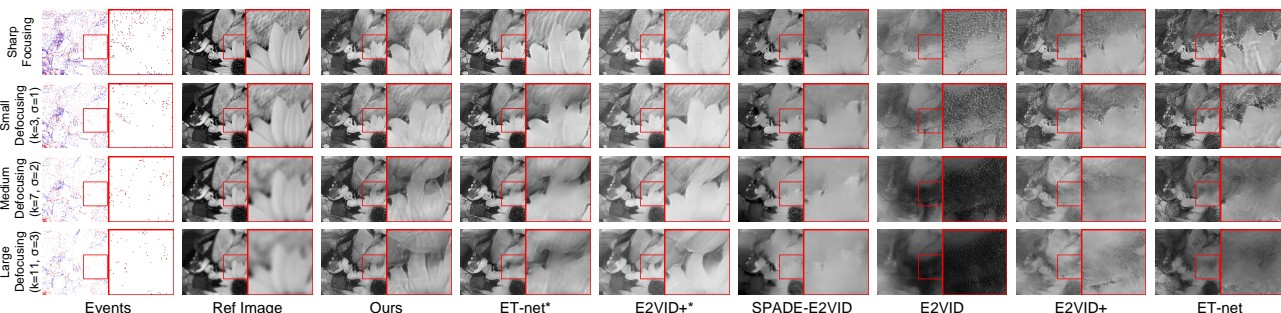

*Figure 5.* Qualitative comparison with baseline methods on out-of-focus simulated data. Our reconstruction result has the best quality, is closest to the sharp ground truth, and restores more details.

that the blur decoder focuses solely on understanding and correcting defocus blur, while the alignment decoder works on aligning features correctly for the final sharp image reconstruction. The spatio-temporally encoded features are processed in parallel through two decoders, namely the blur feature decoder $D_{\text{blur}}$ and the aligned feature decoder $D_{\text{align}}$, generating the blur feature $F_{\text{blur}}$ and the aligned feature $F_{\text{align}}$. Each decoder consists of three decoder blocks, where each decoder block is composed of bilinear upsampling and a convolutional block with a kernel size of 5, followed by ReLU and BN.

The features generated by the blur feature decoder $D_{\text{blur}}$ undergo a prediction layer and are then used to learn blur reconstruction with the blur ground truth (blur gt), which subsequently supervises the redegradation process. The aligned feature decoder $D_{\text{align}}$ performs preliminary alignment, followed by deblur reconstruction and the redegradation process. After each decoder block, the Cross-Modal Fusion (CF) applies ReLU activation to the two features to assist in extracting alignment features, and then concatenates them along the channel dimension. Through the CF module, information from both parts is combined, enhancing the features of the alignment decoder module, thus better focusing on extracting alignment information.

**Reconstruction and Re-defocus Module.** We design a self-supervised loss mechanism where the aligned feature $F_{\text{align}}$ first passes through a Re-defocus module and a Reconstruction module. The Re-defocus module generates a Re-defocus Image, which is used in self-supervised learning with the blur feature $F_{\text{blur}}$, allowing the model to better learn defocus blur distribution in the Blur-aware Two-branch Decoder. The aligned feature $F_{\text{align}}$ is then processed by the Reconstruction module to produce the final clear image reconstruction under temporal consistency supervision.

Our Reconstruction and Re-defocus Module utilizes Transformer-based Multi-Head Channel Attention (MHCA)s (Vaswani, 2017; Zhang et al., 2024). to enhance defocus blur handling. Transformers are selected for their

ability to focus on critical features over long distances, addressing the quadratic growth issue of key-query interactions by applying self-attention across channels and computing cross-channel covariance for generating the attention map. Given queries (Q), keys (K), and values (V), we reshape Q and K such that their dot product generates a transposed attention map $\mathbf{A} \in \mathbb{R}^{C \times C}$ instead of the traditional $\mathbb{R}^{HW \times HW}$. Overall, MHCA can be summarized as:

$$\mathbf{X}' = \mathbf{W_p} \, \text{Attention}(\mathbf{Q}, \mathbf{K}, \mathbf{V}) + \mathbf{X}, \qquad (8)$$

$$\text{Attention}(\mathbf{Q}, \mathbf{K}, \mathbf{V}) = \mathbf{V} \cdot \text{softmax}\left(\frac{\mathbf{K} \cdot \mathbf{Q}}{\alpha}\right), \quad (9)$$

where $\mathbf{X}'$ and $\mathbf{X}$ are input and output feature maps, $\mathbf{W_p}$ is the $1 \times 1$ point-wise convolution, and $\alpha$ is a learnable scaling parameter to control the magnitude of $(\mathbf{K} \cdot \mathbf{Q})$ before applying softmax.

**Self-supervised Mechanism** We design a self-supervised loss mechanism. First, the aligned feature $F_{\text{align}}$ passes through a Re-defocus module and a Reconstruction module. The Re-defocus module outputs a Re-defocus Image, which undergoes self-supervised learning with the blur feature $F_{\text{blur}}$, allowing the model to further learn the distribution of defocus blur in the Blur-aware Two-branch Decoder. Finally, the aligned feature $F_{\text{align}}$ passes through the Reconstruction module, and under temporal consistency supervision, the model produces the final clear image reconstruction.

**Loss Function.** As shown in Fig. 4, we use perceptual loss (Zhang et al., 2018) and temporal consistency loss (Lai et al., 2018; Rebecq et al., 2019) to ensure the quality of reconstruction, and we learn the deblurring process through $L_1$ loss. The final loss is the sum of the losses:

$$L_K = \lambda_1 LR_k^1 + \lambda_2 LR_k^2 + \lambda_3 L_k^{TCP} + \lambda_4 L_1, \quad (10)$$

where $L_K^{TCP}$ represents the Temporal Consistency loss, and $LR_k^1$ and $LR_k^2$ respectively represent the perceptual loss of the align feature $F_{align}$ after prediction supervised by the sharp ground truth (sharp gt) and the blur feature $F_{blur}$ after prediction supervised by the defocus ground truth (blur gt). $\lambda_{1,2,3,4}$ is separately set to 4, 1, 7, 1, respectively.

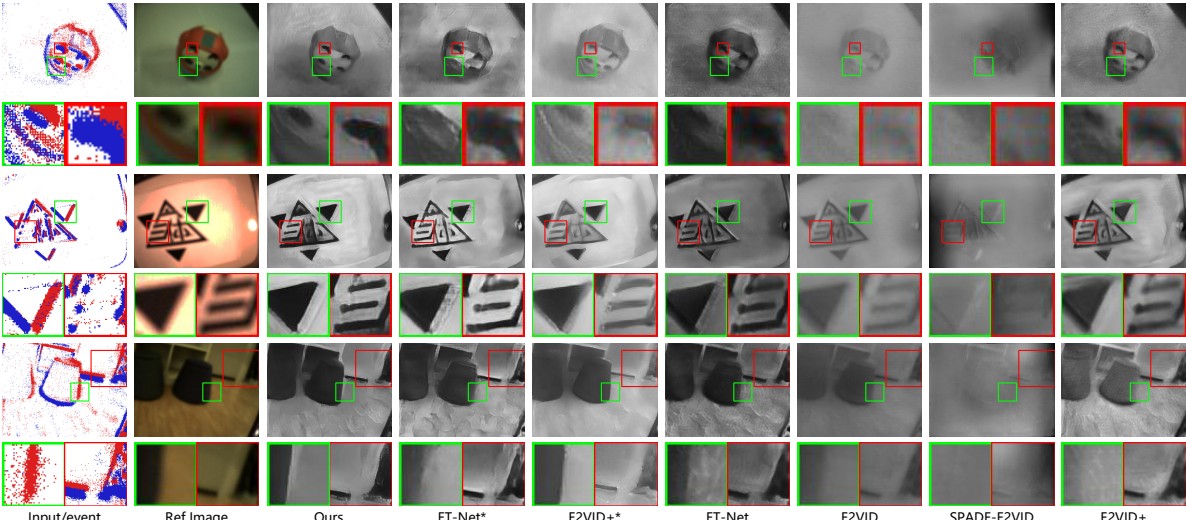

| Input/event | Ref Image | Ours | ET-Net* | E2VID+* | ET-Net | E2VID | SPADE-E2VID | E2VID+ |

*Figure 6.* Qualitative comparison on out-of-focus real-world data. Our proposed model achieves better reconstruction results with the clear edge restored.

*Table 1.* Quantitative comparison on simulated event data with different defocus degrees.

|  | Metric | sharp | k=3, $\sigma$=1.0 | k=5, $\sigma$=1.5 | k=7, $\sigma$=2.0 | k=9, $\sigma$=2.5 | k=11, $\sigma$=3.0 | Mean |
|---|---|---|---|---|---|---|---|---|
| **MSE ↓** | E2VID | 0.084 | 0.065 | 0.065 | 0.071 | 0.080 | 0.085 | 0.075 |
|  | E2VID+ | 0.041 | 0.044 | 0.054 | 0.061 | 0.065 | 0.065 | 0.055 |
|  | E2VID+* | 0.054 | 0.047 | 0.047 | 0.047 | 0.050 | 0.052 | 0.050 |
|  | ET-Net | 0.037 | 0.037 | 0.042 | 0.044 | 0.047 | 0.048 | 0.043 |
|  | ET-Net* | 0.032 | 0.029 | **0.027** | 0.027 | **0.027** | **0.028** | 0.028 |
|  | **Ours** | **0.030** | **0.028** | 0.027 | **0.026** | **0.027** | 0.028 | **0.027** |
| **SSIM ↑** | E2VID | 0.256 | 0.234 | 0.216 | 0.202 | 0.190 | 0.184 | 0.214 |
|  | E2VID+ | 0.342 | 0.309 | 0.282 | 0.265 | 0.248 | 0.237 | 0.281 |
|  | E2VID+* | 0.497 | 0.486 | 0.450 | 0.417 | 0.385 | 0.360 | 0.433 |
|  | ET-Net | 0.352 | 0.331 | 0.305 | 0.283 | 0.261 | 0.247 | 0.297 |
|  | ET-Net* | 0.425 | 0.428 | 0.413 | 0.392 | 0.367 | 0.349 | 0.396 |
|  | **Ours** | **0.550** | **0.525** | **0.486** | **0.449** | **0.413** | **0.385** | **0.468** |
| **LPIPS ↓** | E2VID | 0.344 | 0.361 | 0.379 | 0.394 | 0.406 | 0.422 | 0.384 |
|  | E2VID+ | 0.234 | 0.258 | 0.279 | 0.295 | 0.309 | 0.316 | 0.282 |
|  | E2VID+* | 0.173 | 0.173 | 0.188 | 0.200 | 0.213 | 0.226 | 0.196 |
|  | ET-Net | 0.213 | 0.234 | 0.254 | 0.273 | 0.289 | 0.303 | 0.261 |
|  | ET-Net* | 0.151 | 0.155 | 0.168 | 0.180 | 0.192 | 0.204 | 0.175 |
|  | **Ours** | **0.144** | **0.146** | **0.161** | **0.169** | **0.182** | **0.195** | **0.166** |

## 5. Experiment

**Experimental Setup.** In this section, we compare our method with mainstream reconstruction techniques both qualitatively and quantitatively using simulated and real data. We test our model on simulated data with varying blur levels to demonstrate its adaptability to different degrees of defocus blur, and on real data with varying motion speeds and different focal planes to show its effectiveness in handling motion blur and robustness across varying depth levels. And we visualize the feature in our model to demonstrate the effectiveness of the model architecture. Additionally, we perform ablation studies to evaluate the contribution of each module in our proposed method. **1) Dataset.** As stated in Sec. 2, we generate sequences of defocus events, sharp images, and optical flows, in which 41 sequences are used in the training set and 6 sequences in the test set. To verify the effectiveness of our model on real data, we use the DAVIS 346 cameras to capture 7 real-world

scenes. **2) Implement Details.** Our model is implemented using the PyTorch framework. We adopt a constant strategy of learning rate during training, which is set at 1e-4. Our model is trained for 300 epochs with batch size of 1 on 3 NVIDIA GeForce RTX 3090 GPUs. **3) Evaluation metrics.** For quantitative evaluation on synthetic data, we consider three widely-used evaluation metrics: (i) structure similarity (SSIM) (Wang et al., 2004),(ii) mean squared error (MSE), and (iii) perceptual similarity (LPIPS) (Zhang et al., 2018). For quantitative evaluation on real data,we use two No-Reference Image Quality Assessment metrics: Brisque (Mittal et al., 2012a) and Niqe (Mittal et al., 2012b). BRISQUE (Blind/Referenceless Image Spatial Quality Evaluator) evaluates natural scene statistics in the spatial domain without requiring a reference image, while NIQE (Natural Image Quality Evaluator) computes quality based on deviations from learned statistical regularities of natural images.

**Evaluation on Simulated Event Data. 1) Qualitative Re-**

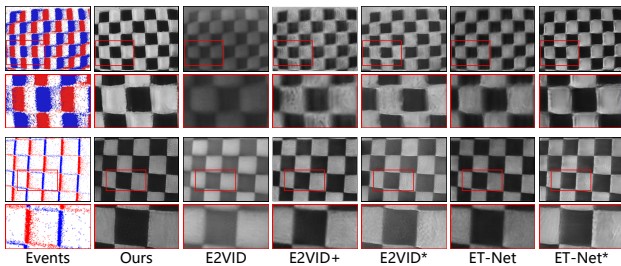

*Figure 7.* Qualitative comparison with baseline methods on real data with different speeds. Our proposed model has better results with sharp details and fewer artifacts.

*Table 2.* Quantitative comparisons on the real-world data.

| Model | Brisque↓ | Niqe↓ |
|---|---|---|
| **Ours** | **27.136** | **6.345** |
| E2VID | 30.977 | 12.505 |
| E2VID+ | 38.957 | 7.986 |
| E2VID+* | 34.399 | 6.877 |
| ET-Net | 31.994 | 7.771 |
| ET-Net* | 31.855 | 6.868 |

**sults.** Fig. 5 presents the quantitative results obtained by evaluating our method and current mainstream methods on the simulated data sequence generated by us. E2VID+* and ET-Net* denote the models are fine-tuned on our dataset. Among them, the first column represents the reconstruction results of clear events and sharp ground truth. The other columns represent different degrees of blur. The larger the kernel size and the larger the $\sigma$, the bigger the blur. Each row shows the reconstruction result of each method for different blurred events. The last row gives the ground truth as a reference. As shown in Fig. 6, E2VID (Rebecq et al., 2019) and E2VID followed by MIMO-UNet or MIMO-UNet+ (Cho et al., 2021b) have the lowest quality and the most prominent artifacts and blurs, and are basically unrecognizable. SPADE-E2VID (Cadena et al., 2021) has relatively fewer artifacts, but the degree of blur is still quite large. E2VID+ (Stoffregen et al., 2020) and ET-Net (Weng et al., 2021) have relatively less blur than the former. E2VID+ and ET-Net retrained on the defocus synthetic data further reduce blur and distortion. However, the contrast of E2VID+ and the retrained E2VID+ still has serious distortion. And our results are also better than all the above results and are the closest to the image quality of the sharp ground truth. **2) Quantitative Results.** We calculate the average value of each metric on all frames in different blur degree groups as in Table 1. The EvFocus method achieves state-of-the-art performance in all metrics. These results validate the effectiveness of the proposed EvFocus method, and the reconstructed images generated by this method are more perceptually satisfactory and high-fidelity.

**Evaluation on Real-world Event Data. 1) Qualitative Results.** We present the qualitative results in Fig. 6, in which each row represents the reconstruction result of the data captured in one scene. The first and second columns respec-

*Table 3.* Ablation study.

| Model | SSIM↑ | MSE↓ | LPIPS↓ |
|---|---|---|---|
| **All Modules** | **0.604** | **0.021** | **0.139** |
| w/o CF | 0.603 | 0.021 | 0.141 |
| w/o sharp supervision ($F_{align}$) | 0.577 | 0.024 | 0.141 |
| w/o $L_1$ re-defocus supervision | 0.403 | 0.021 | 0.140 |
| w/o Reconstruction Module | 0.601 | 0.022 | 0.139 |

tively represent the Events view and the original APS frame captured by the camera. Each of the remaining columns represents the reconstruction result of one method. Fig. 6 shows SPADE-E2VID has the worst effect and the greatest degree of blur. E2VID can reconstruct basic objects, but the degree of blur is still quite large. E2VID+ and ET-Net have better reconstruction, but still remain a lot of artifacts and blurs. The retrained E2VID+ can eliminate blur to a certain extent, but high-frequency information such as edges still cannot be presented well. The retrained ET-Net can reconstruct high-frequency information to a certain extent. Compared with the above results, our method has the best reconstruction quality, and the restoration of details such as edges is the closest to ground truth. **2) Quantitative Results.** Since the visual results of SPADE-E2VID are poor, and the image deblurring of MIMO-UNet or MIMO-UNet+ connected after E2VID does not improve the quality of the reconstructed image, we only conduct quantitative comparisons of the remaining methods. Among them, our method achieves the best performance. We present quantitative results on real data in Table 2. We omit SPADE-E2VID and E2VID followed by MIMO-UNet or MIMO-UNet+ due to poor quantitative scores. Our method achieves the best results on both Brisque and Niqe metrics.

**Ablation Study.** To highlight the effectiveness of each module in our model, we conduct ablation experiments as shown in Table 3. As illustrated in the table:

**1) Effect of sharp supervision to $F_{align}$.** Removing the perceptual loss supervised by the sharp ground truth results in the worst performance in MSE and LPIPS. This demonstrates that the align decoder plays a crucial role in the reconstruction and re-degradation process by providing prior knowledge to align $F_{align}$ with the followed reconstruction. **2) Effect of Reconstruction Module.** The model without the reconstruction module also shows worse performance, indicating that the reconstruction module, as the last module before producing the clear predicted image, improves the quality of the reconstruction results. **3) Effect of CF.** Removing the CF structure leads to a decline in the model's performance, proving the effectiveness of our strategy to enhance aligned features through Cross-Modal Fusion. **4) Effect of $L_1$ re-defocus supervision.** The model without the $L_1$ loss also shows a noticeable decrease, further validating the importance of our re-degradation module.

**Feature Visualization.** We visualize the intermediate fea-

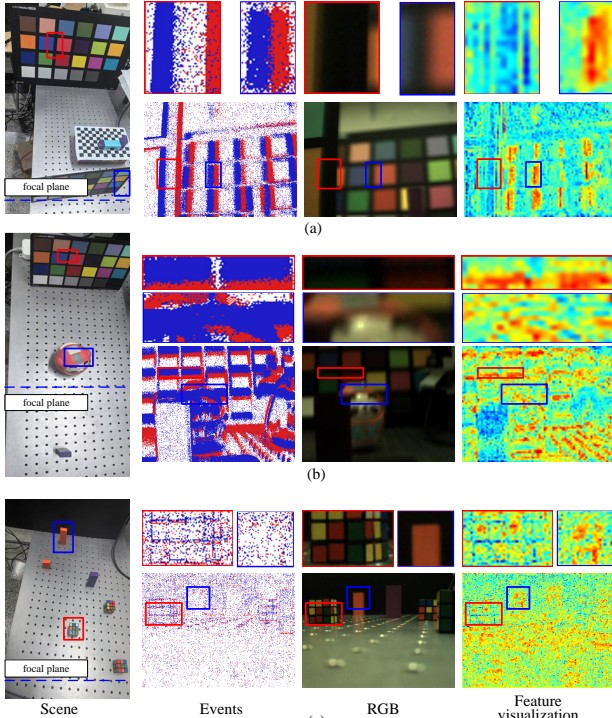

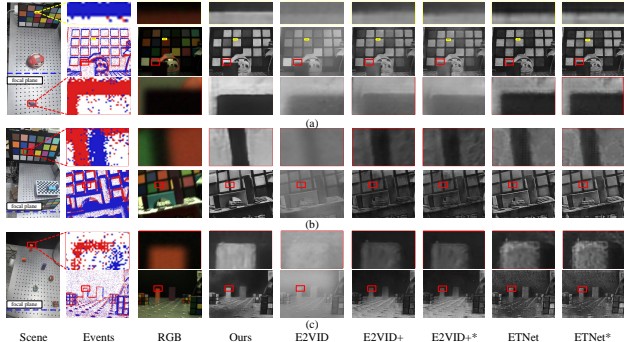

Figure 9. Qualitative experiment on different depth levels. We capture three scenes with distinct depth variations for evaluation. Our method demonstrates effective deblurring performance across different depth levels.

*Figure 8.* Feature visualization. We visualize the features input to the decoder. Taking (a) as an example, the red box indicates a region with slight defocus, while the blue box shows a region with severe defocus. Despite similar event densities caused by motion, the feature map reveals stronger responses in the heavily blurred area. This demonstrates that our network exhibits varying feature intensities for blur at different depths, validating the model's physical consistency.

tures of our model using real-world data containing scenes with objects at varying degrees of defocus in Fig. 8. The results demonstrate that our model can effectively perceive features across different defocus levels, validating the effectiveness of the model design.

**Evaluation on Different Out-of-focus Blur Size.** In Fig. 5, we conduct experiments on the degree of blur that our model can handle. Through experiments, our model can perform relatively complete and clear reconstructions for blurred events with kernel size less than or equal to 7. For blurred events with kernel size greater than 7, the reconstruction quality of our model can also be the best.

**Evaluation on Different Depth Level.** We conduct experiments on scenes with varying depths to validate our model's ability to recover texture information at different focal lengths within the same scene as in Fig. 9. The experimental results demonstrate that our model achieves effective defocus recovery across different focal settings.

**Evaluation on Different Motion Speed.** To test the processing effect of our model on different blurs, we collect real event data at different moving speeds for experiments

(Fig. 7). E2VID has only slight motion blur at low speed, but does not effectively remove the motion blur at high speed. The reconstruction result is generally blurry. E2VID+ and the retrained E2VID+ have better reconstruction quality at low speed, but cannot effectively remove high-speed motion blur. ET-Net and the retrained ET-Net have better detail recovery at low speed. The retrained ET-Net has less motion blur at high speeds than ET-Net but still cannot be removed well. In contrast, our model shows better reconstruction quality at both high and low speed, and retains the best details, proving that our method is also superior to the above methods for the reconstruction of different event densities.

**Parameters.** We present the parameter size of each model and inference speed in Table 4. Our model is in the same order of magnitude as mainstream models such as E2VID and is lower than ETNet.

*Table 4.* Comparison on Parameters and Runtime.

| Methods | Params(M) | Inference Time(ms) |
|---|---|---|
| E2VID | 10.71 | 4.41 |
| ETNet | 22.18 | 25.71 |
| SPADE | 11.46 | 12.66 |
| **EvFocus (ours)** | 12.11 | 24.09 |

# 6. Conclusion

We propose EvFocus, a novel framework for reconstructing sharp images from defocus event streams. It incorporates a temporal encoder and a blur-aware decoder to effectively correct defocus blur. Additionally, we develop a defocus event simulator to generate realistic training data, enabling robust model performance. Unlike traditional methods focused on spatial restoration, EvFocus leverages the temporal characteristics of event streams for improved defocus handling. Experiments on simulated and real-world datasets show that EvFocus outperforms existing methods, demonstrating its effectiveness in event-based defocus imaging.

## Impact Statement

This paper presents work whose goal is to advance the field of Machine Learning. There are many potential societal consequences of our work, none which we feel must be specifically highlighted here.

## Acknowledgments

This work is partially supported by National Natural Science Foundation of China under Grant No.62302041 and China National Postdoctoral Program under contract No.BX20230469.

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

# A. Appendix: Theoretical Framework for Simulation

Our motivation is to explore the temporal characteristics of defocus-induced event streams, which are crucial for understanding the dynamic interactions between motion, scene structure, and defocus effects. Unlike traditional defocus blur datasets, which primarily focus on the spatial restoration of sharp images from blurred ones, our work emphasizes the generation and analysis of event streams in the temporal domain. This shift in focus allows us to model and study how defocus blur impacts the asynchronous nature of event data, capturing fine-grained temporal dynamics that are often overlooked in conventional datasets. By incorporating motion dynamics and adopting multiple strategies, we aim to simulate real-world scenarios with greater fidelity and provide new insights into event-based vision under defocus conditions.

## A.1. Event Triggering Model and Intensity Dynamics

Event cameras capture temporal intensity changes at a high temporal resolution, generating asynchronous events based on logarithmic intensity differences. The triggering condition for an event at pixel $(x, y)$ and time $t$ can be formalized as:

$$E(x, y, t) = \begin{cases} 1, & \text{if } \frac{\Delta L(x,y,t)}{L(x,y,t)} \geq C_+, \\ -1, & \text{if } \frac{\Delta L(x,y,t)}{L(x,y,t)} \leq C_-, \\ 0, & \text{otherwise,} \end{cases} \tag{11}$$

where:

- $L(x, y, t)$ denotes the light intensity at pixel $(x, y)$ at time $t$,

- $\Delta L(x, y, t) = L(x, y, t) - L(x, y, t - \Delta t)$ represents the temporal change in intensity over a time interval $\Delta t$,

- $C_+$ and $C_-$ are predefined positive and negative contrast thresholds, respectively.

The magnitude of $\Delta L(x, y, t)$ denotes temporal intensity gradient, which is influenced by two critical factors:

1. The *relative motion* between the camera and the observed scene, which induces pixel-level intensity shifts in the image plane.

2. The *spatial intensity gradient*, $\nabla L(x, y)$, representing local intensity variations.

These factors collectively determine the event triggering behavior and the density of the resulting event stream.

## A.2. Impact of Blur on Intensity Gradients

Optical blur, such as defocus, attenuates spatial intensity variations by smoothing local intensity gradients, effectively reducing $\nabla L(x, y)$. This smoothing effect can be mathematically represented using a Gaussian blur kernel $G_\sigma$, defined as:

$$L_{\text{blur}}(x, y, t) = G_\sigma * L(x, y, t), \tag{12}$$

where $G_\sigma$ is a Gaussian kernel with standard deviation $\sigma$ that determines the blur magnitude. Consequently, the blurred spatial intensity gradient is expressed as:

$$\nabla L_{\text{blur}}(x, y, t) = G_\sigma * \nabla L(x, y, t). \tag{13}$$

As the blur strength ($\sigma$) increases:

1. The spatial intensity gradient $\nabla L_{\text{blur}}(x, y, t)$ diminishes, leading to a reduced magnitude of temporal intensity changes, $\Delta L(x, y, t)$.

2. The reduction in intensity changes results in fewer events being generated, producing a sparser event stream.

Thus, the relationship between blur and event sparsity forms a fundamental aspect of modeling the effect of optical defocus on event generation.

### A.3. Motion-Induced Intensity Changes

Relative motion between the camera and the scene is another primary source of temporal intensity variation. Let the motion-induced speed in the image plane be $V_p$. The temporal change in intensity due to motion can be approximated as:

$$\Delta L(x, y, t) \propto V_p \cdot \nabla L_{\text{blur}}(x, y, t), \tag{14}$$

where:

- $V_p$ is the motion velocity projected onto the image plane,

- $\nabla L_{\text{blur}}(x, y, t)$ is the blurred spatial gradient at pixel $(x, y)$.

From this expression, it is evident that:

- **High motion velocity** ($V_p$ large) amplifies temporal intensity changes, thereby mitigating the smoothing effects of blur.

- **Low motion velocity** ($V_p$ small) reduces intensity changes, making the impact of blur more pronounced.

This interaction highlights the interplay between motion dynamics and optical blur in determining event density and spatial distribution.

### A.4. Interdependence of Motion Speed and Blur Strength

If an object moves with velocity $V_p$, we can simulate the object's movement by applying geometric transformations $T(t)$ (such as translation, rotation, or scaling):

$$L_{\text{transformed}}(x, y, t) = T(t)L(x, y, 0). \tag{15}$$

Event triggering depends on the change in light intensity, so:

$$\Delta L(x, y, t) = L_{\text{transformed}}(x, y, t) - L_{\text{transformed}}(x, y, t - \Delta t). \tag{16}$$

If we choose an appropriate motion velocity $V_p$, the intensity change rate generated by the geometric transformation matches that produced by different depth-related defocus blur, as shown by:

$$\frac{\Delta L(x, y, t)}{L(x, y, t)} \approx \frac{\Delta L_{\text{blur}}(x, y, t, d)}{L_{\text{blur}}(x, y, t, d)}. \tag{17}$$

Thus, for a given motion speed $V_p$, the events triggered by motion can be equivalent to those triggered by depth-related defocus blur.

**Theorem 1 (Equivalence Theorem)**: There exists a mapping between the blur level $\sigma(d)$ and the motion speed $V_p$ such that for each depth $d$, the event density generated by motion at a specific velocity are equivalent to those generated by defocus blur at that depth:

$$\sigma(d) \propto \frac{1}{V_p}. \tag{18}$$

The preceding analysis suggests an inverse relationship between motion speed ($V_p$) and defocus blur strength ($\sigma$) in their influence on temporal intensity changes and event generation:

$$\Delta L(x, y, t) \propto V_p \cdot \nabla L_{\text{blur}}(x, y, t). \tag{19}$$

This relationship implies that:

1. High-speed motion ($V_p$ large) can counteract the effect of a small blur kernel ($\sigma$ small).

2. Low-speed motion ($V_p$ small) amplifies the impact of a large blur kernel ($\sigma$ large).

This equivalence facilitates the simulation of varying levels of blur by appropriately adjusting the motion speed.

### A.5. Simulation Pipeline

The analysis demonstrates that the generation of defocus-induced event streams is influenced by motion speed, scene intensity, and the degree of defocus. Unlike traditional defocus-related work that focuses on image restoration in the spatial domain, our research primarily investigates the temporal characteristics of defocus event streams. To model this, we adopt different backgrounds and foregrounds, complex motion trajectories and motion parameters, and multiple Gaussian blur kernels with varying standard deviations $\sigma$, the simulation process is structured as follows:

---

**Algorithm 1** Synthetic Data Generation Pipeline

---

**Require:** • A set of background images $\{\mathcal{B}_i\}$

    • A set of foreground images $\{\mathcal{F}_{i,j}\}$ for each background $i$

    • Motion parameters (e.g. translation, rotation) for background & foreground

    • Number of time steps $T$

**Ensure:** • Synthetic dataset containing rendered scenes with events & optical flow

1: **for** each scene $i$ **do**
2:    Select one background image $\mathcal{B}_i$
3:    Select $M$ foreground images $\{\mathcal{F}_{i,j}\}_{j=1}^{M}$
4:    Generate motion trajectories for background and each foreground:

       • $\mathbf{traj}_{\mathcal{B}} \leftarrow$ GenerateTrajectory(motion parameters)

       • $\mathbf{traj}_{\mathcal{F}_{i,j}} \leftarrow$ GenerateTrajectory(motion parameters)

5:    **for** $t = 1, \ldots, T$ **do**
6:      Sample camera pose $\mathbf{p}_t \leftarrow$ SamplePose()
7:      Sample camera distortion $\mathbf{d}_t \leftarrow$ SampleDistortion()
8:      Render defocus brightness image:

$$I_t \leftarrow \text{Render}(\mathcal{B}_i, \{\mathcal{F}_{i,j}\}, \mathbf{traj}_{\mathcal{B}}[t], \{\mathbf{traj}_{\mathcal{F}_{i,j}}[t]\}, \mathbf{p}_t, \mathbf{d}_t)$$

9:      Compute brightness change $\Delta I_t = I_t - I_{t-1}$ (if $t > 1$)
10:     Generate events $\mathcal{E}_t \leftarrow$ EventGeneration($\Delta I_t$)
11:     Compute optical flow $\mathbf{u}_t \leftarrow$ OpticalFlow($I_t$)
12:    **end for**
13:    Store $\{I_t\}_{t=1}^{T}$, $\{\mathcal{E}_t\}_{t=1}^{T}$, $\{\mathbf{u}_t\}_{t=1}^{T}$ as the dataset for scene $i$
14: **end for**

---

# B. Appendix: Network Architecture Details

Fig. 4 provides an overview of the EvFocus network, as detailed in Section 3. In this Section, we will explain other details of the model.

## B.1. Temporal Information Encoder.

Our temporal information encoder module consists of a head layer and three encoder modules. The head layer outputs a number of channels $N_b = 32$, and the number of channels is doubled after each encoder layer. The inputs to our temporal information encoding module are 5-channel voxel-grids, and the outputs are feature maps with $N_b \times 2^3$ channels.

## B.2. Self-supervised Mechanism

We design a self-supervised loss mechanism. First, the aligned feature $F_{\text{align}}$ passes through a Re-defocus module and a Reconstruction module. The Re-defocus module outputs a Re-defocus Image, which undergoes self-supervised learning with the blur feature $F_{\text{blur}}$, allowing the model to further learn the distribution of defocus blur in the Blur-aware Two-branch Decoder. Finally, the aligned feature $F_{\text{align}}$ passes through the Reconstruction module, and under temporal consistency supervision, the model produces the final clear image reconstruction.

## B.3. Reconstruction and Re-defocus Module

The Reconstruction and Re-defocus Module is designed to refine the network's ability to handle defocus blur. By utilizing Transformer-based Multi-Head Channel Attention (MHCA), we aim to improve the network's focus on the most relevant features for blur correction. Transformers are chosen because they offer the flexibility to attend to important features over long distances in the feature space, which is essential for effectively capturing the intricate patterns of defocus blur.

Our Reconstruction and Re-defocus Module consists of sequentially cascaded Transformer-based Multi-Head Channel Attention (MHCA) modules (Vaswani, 2017; Zhang et al., 2024). Transformers have demonstrated unique advantages in image restoration tasks due to their flexible and adaptive kernels. To address the quadratic growth problem of key-query dot product interactions, we adopt the idea of applying self-attention across channels rather than spatial dimensions and compute the cross-channel covariance to generate the attention map.

## B.4. Loss Fuction

**Perceptual loss** We use the calibrated perceptual loss LPIPS (Zhang et al., 2018) to supervise the reconstruction image $\hat{I}$ with sharp ground truth and two decoder outputs $I_1$ and $I_2$ after processing the prediction with out-of-focus and focused ground truth. The LPIPS loss can evaluate the perceptual similarity of frame quality in a more realistic manner. The perceptual reconstruction loss is specifically calculated as

$$LR_k = d(\hat{I}_k, I_k), \tag{20}$$

where $d$ represents the LPIPS distance.

**Temporal Consistency and Perceptual Joint Loss** To supervise the final reconstructed image, we use a joint loss of temporal consistency loss and perceptual loss. The perceptual loss between the predicted image and the sharp ground truth is used to ensure reconstruction quality. At the same time, due to the presence of temporal artifacts, we use a temporal consistency loss to correct the warping error between the reconstructed consecutive frames through the optical flow map (Lai et al., 2018). This is specifically expressed as:

$$L_k^{TC} = M_{k-1}^k \left\| \hat{I}_k - W_{k-1}^k(\hat{I}_{k-1}) \right\|_1, \tag{21}$$

where $d$ represents the LPIPS distance (Zhang et al., 2018), $W_{k-1}^k(\hat{I}_{k-1})$ is the result of warping the reconstruction $\hat{I}_{k-1}$ to $\hat{I}_k$ using the optical flow $F_{k-1}^k$, and $M_{k-1}^k = \exp(-\alpha \left\| I_k - M_{k-1}^k(I_{k-1}) \right\|_2^2)$ is a weighting term that helps mitigate the effects of occlusion.
Therefore, the Temporal Consistency and Perceptual joint Loss can be formulated as:

$$L_K^{TCP} = \lambda LR_k + L_k^{TC}, \tag{22}$$

where $L_K^{TCP}$ represents the Temporal Consistency loss, $LR_k$ represents the perceptual loss, and $\lambda$ is set to 0.75.

**L1 Loss** To ensure that the re-defocus module can learn defocus blur, we use the $L_1$ loss to supervise the output of the re-defocus module with the output of the blur decoder through the prediction layer, which can be described as:

$L_1 = \|\hat{I}_{1k} - \hat{I'}_k\|_1$ where $\hat{I}_{1k}$ represents the predicted output of the blur decoder $D_{\text{blur}}$, and $\hat{I'}_k$ represents the output of the re-defocus module.

**Total Loss**  The final loss is the sum of the losses mentioned above:

$$L_K = \lambda_1 LR_k^1 + \lambda_2 LR_k^2 + \lambda_3 L_k^{TCP} + \lambda_4 L_1, \tag{23}$$

where $LR_k^1$ and $LR_k^2$ respectively represent the perceptual loss of the align feature $F_a lign$ after prediction supervised by the sharp ground truth (sharp gt) and the blur feature $F_b lur$ after prediction supervised by the defocus ground truth (blur gt). $\lambda_{1,2,3,4}$ is separately set to 4, 1, 7, 1, respectively.

### B.5. Dataset Details

To evaluate the applicability of our method, we conduct tests using seven real-world scenes as in Fig. 10. These scenes are captured by DVS 346, where events and frames are $346 \times 260$ resolution. The "Pic" test assesses the reconstruction performance for defocus events with similar focal lengths, while the "motionspeed1" and "motionspeed2" tests evaluate the model's suitability for handling motion blur. The "desk" "board" "library" and "toy" tests examine the reconstruction performance for defocus events in scenes with different blur kernels and significantly different focal lengths.

*Table 5.* Statistics for Test Scenes

| Scene | Number of Frames | Number of Events |
|---|---|---|
| pic | 135 | 7912802 |
| motionspeed1 | 169 | 23476852 |
| motionspeed2 | 41 | 11426136 |
| desk | 289 | 14980786 |
| board | 204 | 36806803 |
| library | 223 | 10912574 |
| toy | 377 | 6575390 |

## C. Appendix: More Experimental Results

In this section, we primarily present the qualitative comparison of our model with several mainstream models across all real-world scenes and the parameter size for each model.

### C.1. Qualitative Comparison

In Fig. 11 and Fig. 12, we compare the reconstruction results of our model with E2VID, E2VID+, ETNET and SPADE E2VID. From these experimental results, it can be observed that the reconstruction result of our model is the most effective in terms of blur removal. It is capable of clearly reconstructing edge details even when there is a large variance in focal length distribution. Additionally, the motion blur removal effect under varying motion speeds is superior to that of other methods.

In Fig. 13 and Fig. 14, we further evaluate the performance of other mainstream reconstruction methods by using their results as input to a deblurring model, NRKNet (Yuhui Quan & Ji, 2023), and then compare the output with our model's results. The experimental results demonstrate that the reconstructed frames followed by the additional defocus deblurring network, still fail to effectively remove the blur. In contrast, our model achieves high reconstruction quality.

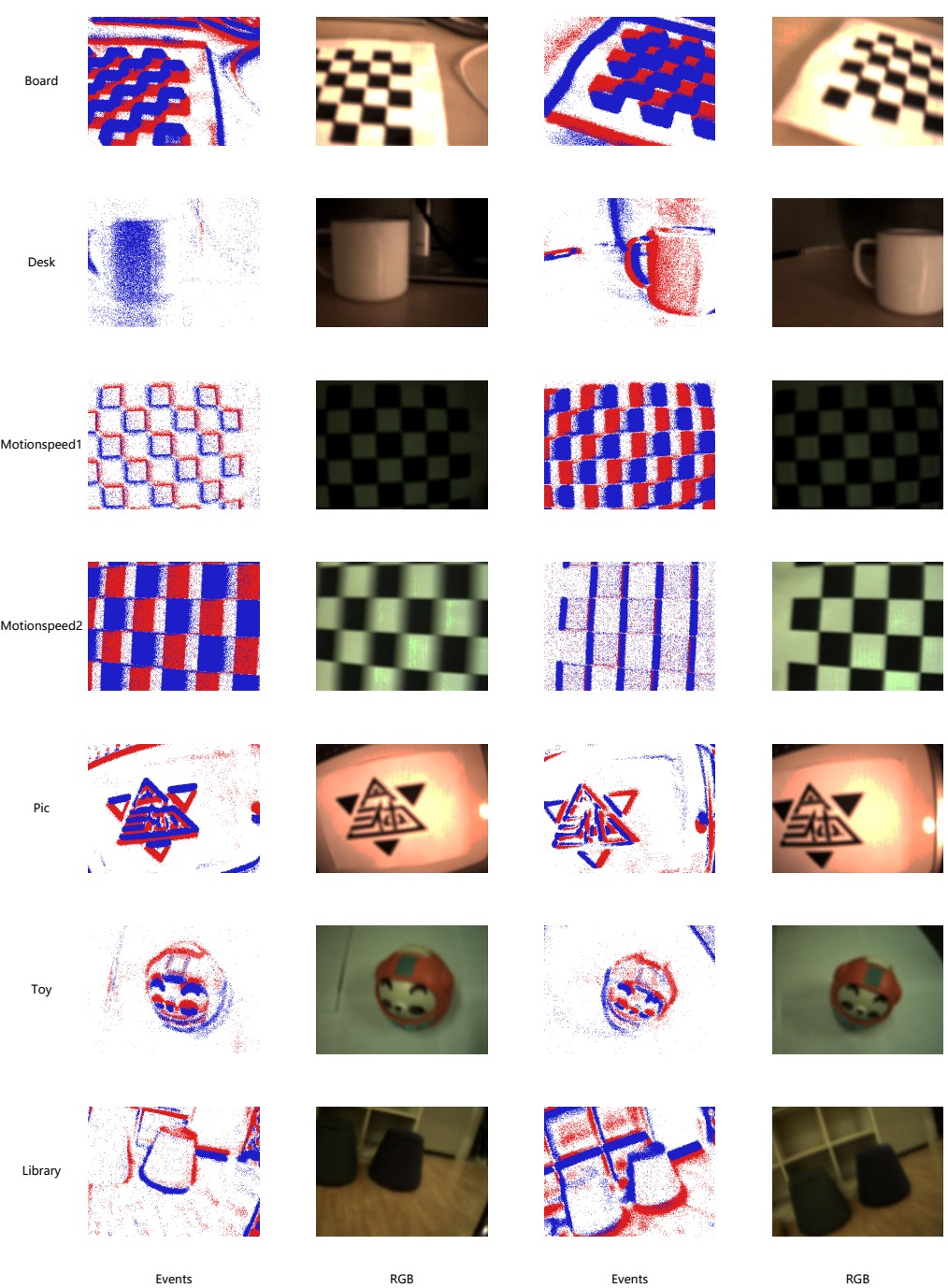

*Figure 10.* Examples of our real-world dataset.

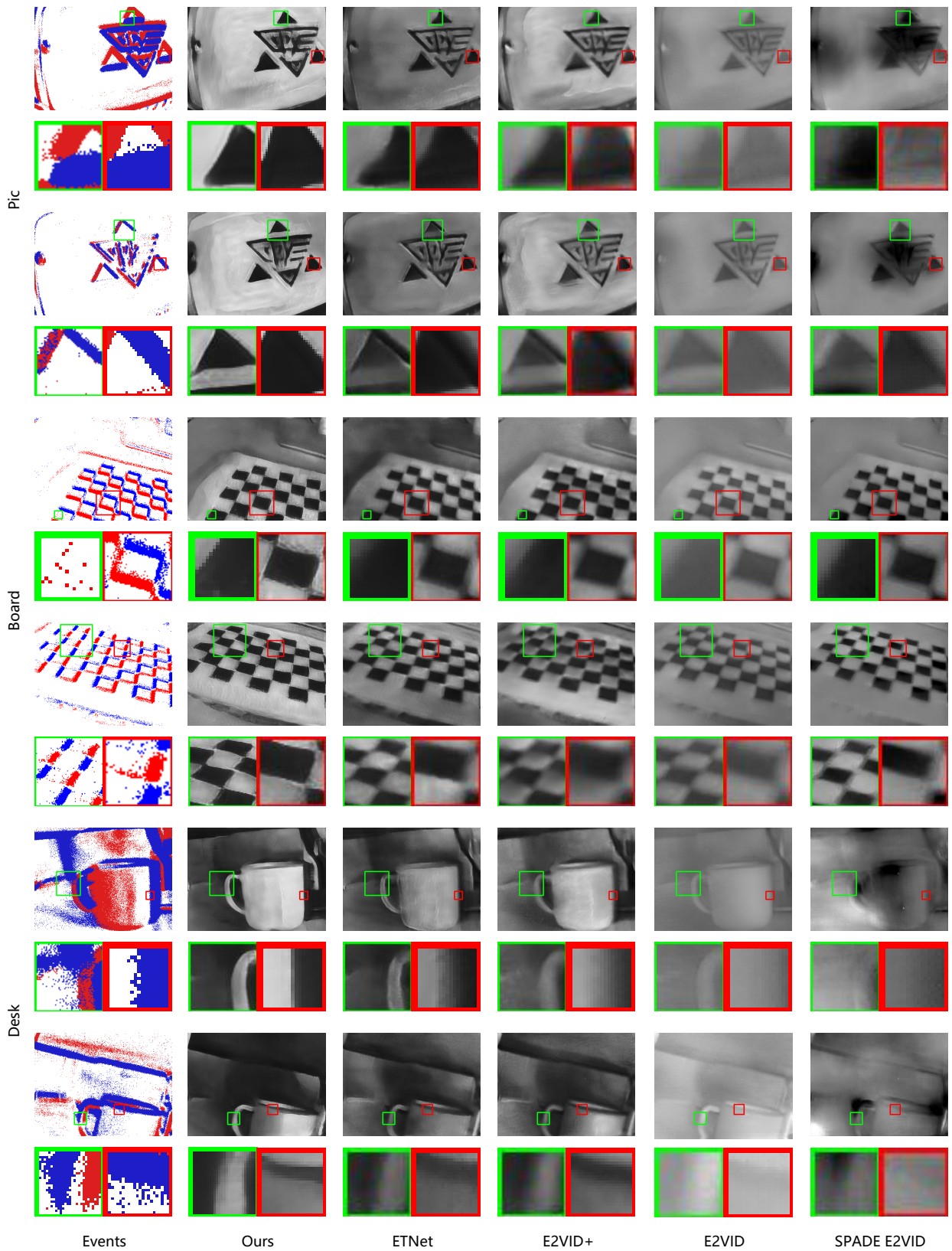

*Figure 11.* Visual comparison results in the Desk, Board, and Pic scenes. We compared our model with E2VID, E2VID+, ETNET and SPADE E2VID.

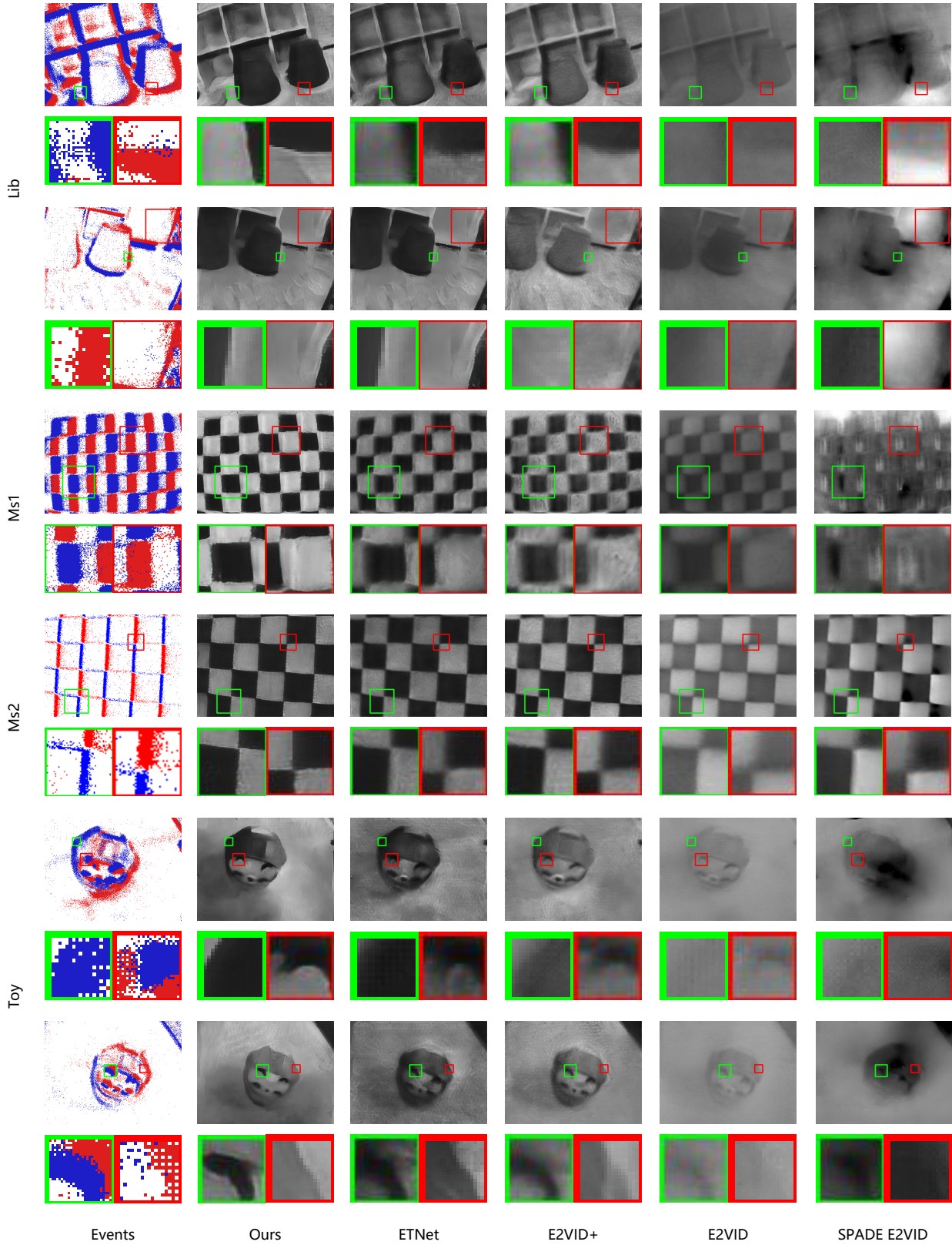

*Figure 12.* Visual comparison results in the Lib, Ms1, Ms2 and Toy scenes. We compared our model with E2VID, E2VID+, ETNET and SPADE E2VID.

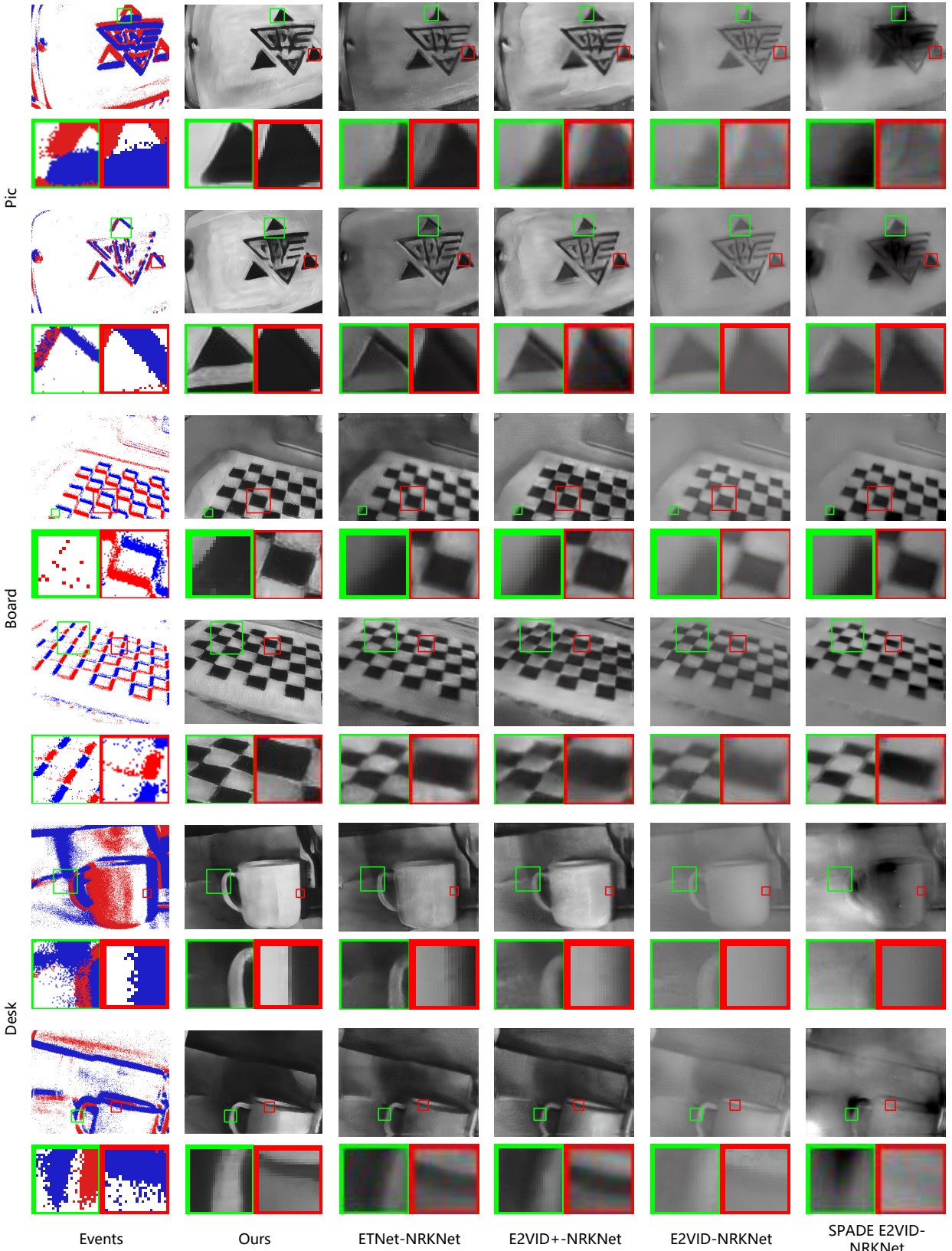

*Figure 13.* Visual comparison of E2VID, E2VID+, ETNET, SPADE E2VID followed by NRKNet and our method in the Pic, Board and Desk scenes. For a better comparison of defocus deblurring, we utilize the reconstruction frames from E2VID, E2VID+, ETNET, and SPADE E2VID as input to NRKNet (Yuhui Quan & Ji, 2023), and compare the results with our method.

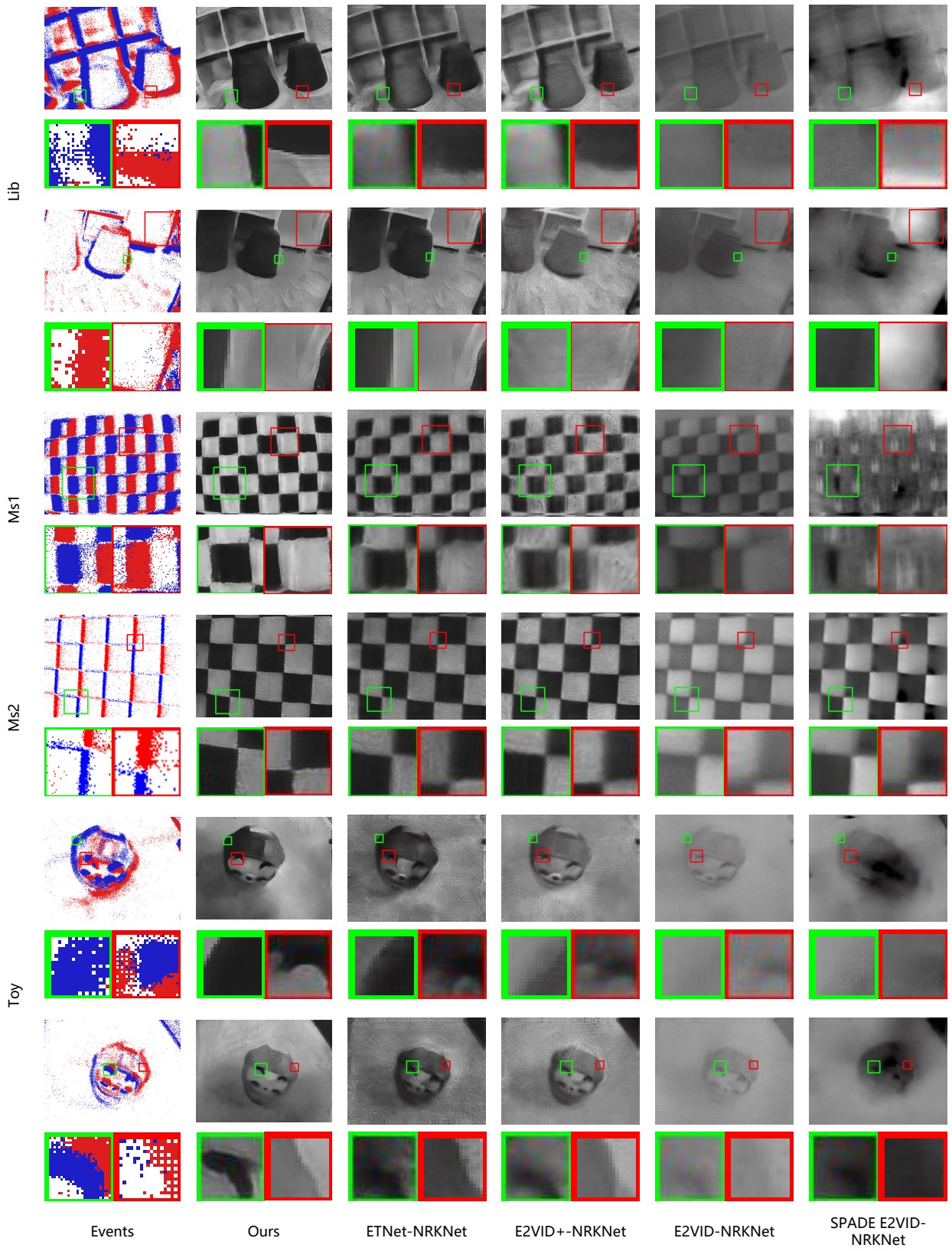

*Figure 14.* Visual comparison of E2VID, E2VID+, ETNET, SPADE E2VID followed by NRKNet and our method in the Lib, Ms1, Ms2 and Toy scenes.

