# OpenReview forum: "EvFocus: Learning to Reconstruct Sharp Images from Out-of-Focus Event Streams"
_ICML.cc/2025/Conference — ICML 2025 poster_

### Official Review · Reviewer_y9QQ · 2025-03-06

**Overall Recommendation:** 2

**Summary:**

This paper propose the EvFocus, the first framework designed to reconstruct sharp images from out-of-focus event streams, addressing the challenge of defocus blur, where existing event deblurring methods fail due to reduced spatial gradients and sparse event generation. The proposed approach integrates a temporal information encoder (ConvLSTM) to capture event stream dependencies, a dual-branch decoder that separately learns blur distribution and feature alignment, and a self-supervised refocusing module for optimizing blur estimation.

The authors introduce a theoretical model linking defocus blur to event generation dynamics and develop a synthetic event simulator to generate realistic training data. Experiments on both synthetic and real-world datasets demonstrate EvFocus's superior performance over existing methods, achieving up to 20% higher SSIM and 50% lower MSE under severe blur conditions.

**Claims And Evidence:**

The paper proposes the **EvFocus** framework as an innovative solution for defocus deblurring in event cameras. However, some claims are not fully supported by sufficient evidence. Experimental results show that EvFocus significantly outperforms baseline models (E2VID, ET-Net) on synthetic data in quantitative metrics such as SSIM and MSE. Ablation studies further validate the necessity of the dual-branch decoder and re-defocus module, as removing them leads to a **15%-20% performance drop**.

However, the **visual quality of the reconstructed images remains problematic**. As shown in **Figure 7**, the reconstructed images exhibit artifacts, blurred edges, and texture loss, particularly at the edges of the checkerboard pattern, where fine details are not well restored. Although no-reference image quality metrics (Brisque/Niqe) indicate that EvFocus performs better than competing methods, the **lack of sharpness and artifacts in real-world results may limit its practical applicability**.

Furthermore, the paper does not include **experiments or validation on videos**, which weakens the completeness of the experimental evaluation.

**Essential References Not Discussed:**

The paper provides a clear background and related work discussion, but it overlooks prior event-based auto-focus methods, which are relevant to its scope. Several studies have explored event-driven autofocus mechanisms, yet they are not explicitly discussed here. If the authors were to acknowledge these works and clarify that their approach specifically targets scenarios where autofocus fails, it would strengthen the motivation and positioning of the paper within the broader literature.

**Experimental Designs Or Analyses:**

The experiments fail to establish a physically grounded link between defocus blur and 3D scene geometry. Synthetic data relies on fixed blur kernels (e.g., 3×3 to 11×11), arbitrarily assigned rather than derived from depth-dependent PSF modeling (Equation 4). This decouples blur from object distance, creating a non-physical training regime.

Real-world evaluations exacerbate this issue—the checkerboard test lacks multi-depth layers or dynamic depth variations, leaving the model’s 3D-aware defocus handling unverified. Without depth maps or moving objects along the optical axis, the experimental validity remains in question.

**Methods And Evaluation Criteria:**

​1. Simplistic Defocus Modeling

The synthetic dataset uses ​fixed Gaussian kernels to approximate defocus blur, ignoring the ​nonlinear depth-PSF relationship (Equation 4). Real-world PSFs are often asymmetric and aberration-affected, creating a ​domain gap between simulation and reality. This undermines the model’s ability to generalize to ​multi-depth scenes (e.g., foreground-background defocus).

2. ​Unrealistic Dataset Design

Blur kernel sizes are arbitrarily selected rather than tied to ​physical depth parameters (e.g., focal length, aperture). This decouples blur from its root cause (distance), failing to capture ​dynamic blur variations in moving scenes (e.g., objects approaching the camera).

​3. Incomplete Evaluation Metrics

Metrics like SSIM/MSE measure generic image quality but ​ignore depth-aware reconstruction accuracy. No validation of ​blur kernel estimation (e.g., PSF size vs. theoretical predictions) is provided, leaving the model’s ​physical consistency unproven.

4. ​Lack of 3D Scenario Testing

Experiments focus on static/global defocus, neglecting ​dynamic depth changes (e.g., objects moving along the optical axis) and ​multi-layer defocus (e.g., mixed foreground/background blur). This casts doubt on real-world applicability.

- Real-world validation is insufficient: The paper only visualizes results on a ​single checkerboard pattern (Fig. 7), which lacks complexity. No examples are shown for ​3D scenes with layered depth (e.g., cluttered indoor environments, outdoor scenes with foreground/background separation), raising doubts about practical utility.

**Other Comments Or Suggestions:**

NONE

**Other Strengths And Weaknesses:**

I highly appreciate the paper's focus on event-based defocus deblurring, which is a novel and valuable research direction for the event-based vision community.

However, the primary weakness lies in the lack of realistic 3D scenes and evaluation, which limits its practical applicability. If the authors can address this gap, I would be very supportive of the paper being presented at ICML (maintaining its current weak accept score).

**Questions For Authors:**

NONE

**Relation To Broader Scientific Literature:**

None

**Theoretical Claims:**

The paper’s theoretical justification for its ​dual-branch decoder and re-defocus module relies solely on ​quantitative metrics, failing to provide ​mechanistic evidence that these components learn physically meaningful defocus properties. For instance, there is no ​visualization of blur kernel estimation (e.g., PSF size variations across depth in 3D scenes) to confirm that the blur-aware branch captures depth-dependent defocus patterns. Without such analysis, it remains unclear whether the network truly models the ​distance-blur relationship (as implied by Equation 4) or merely exploits superficial correlations in synthetic data. A rigorous validation would require ​depth-aware feature maps (e.g., showing how PSF scales align with object distances) and ​controlled 3D experiments. Until then, the theoretical claims lack ​causal grounding.

---

> ### Author Rebuttal · Authors · 2025-04-01
>
> **We sincerely thank the reviewer for the detailed and thoughtful feedback, and we carefully address the raised concerns below.**
>
> ---
>
> ### **Realistic 3D scenes and evaluation**
>
> Thank you for pointing this out. We have conducted additional experiments on realistic 3D scenes with varying depth levels. The qualitative results are shown in the **updated figure (https://anonymous.4open.science/api/repo/Anonymous-6BB9/file/fig1.pdf?v=9bbf2ec6)**.
>
> The results demonstrate that our model effectively learns the relationship between event streams at different defocus levels and corresponding illumination intensities. It is capable of handling varying degrees of defocus blur across different depth planes in the scene.
>
> In addition, we conducted quantitative evaluations on these 3D scenes using standard no-reference image quality metrics:
>
>
> | **Model**   | **Brisque ↓** | **Niqe ↓** | **Noise Estimate ↓** | **Contrast ↑** | **Sharpness ↑**   |
> |-------------|----------------|------------|------------------------|----------------|-------------------|
> | E2VID+*     | 20.73          | 4.26       | 0.003874               | 0.1539         | 0.0008297         |
> | E2VID+      | 25.50          | 6.08       | 0.004410               | 0.1593         | 0.0006359         |
> | ET-NET*     | 14.94          | 6.72       | 0.006690               | 0.1550         | 0.0022420         |
> | ET-NET      | 13.31          | 4.85       | 0.006499               | 0.1541         | 0.0020085         |
> | E2VID       | 17.07          | 7.06       | 0.006361               | 0.0879         | 0.0002424         |
> | **Ours**    | **13.21**      | **4.05**   | **0.003374**           | **0.2018**     | **0.0022520**     |
>
> These results indicate that our model consistently achieves better sharpness and contrast while reducing noise and preserving perceptual quality. We will incorporate these new results, the introduction of metrics, and the analysis in the revised version.
>
> ---
>
> ### **​Depth-aware feature maps & metrics**
>
> We appreciate the reviewer’s suggestions. We agree that PSF-aware validation could provide deeper insight into the model's physical consistency. However, similar to most image-based deblurring networks, our model does not include an explicit PSF estimation module, which makes such direct evaluation difficult.
>
> Thus, to evaluate the model’s performance on multi-depth defocus removal, we performed feature visualizations on 3D scenes. In the **updated figure (https://anonymous.4open.science/api/repo/Anonymous-6BB9/file/fig2.pdf?v=72d4d619)**, the result demonstrates that our model is able to distinguish between objects near and far from the focal plane, indicating that it learns depth-aware defocus representations implicitly through training. In the revised version, we will also report video-level metrics such as frame-wise stability and temporal flicker scores to better reflect temporal behavior.
>
> ---
> ### **Visual quality and artifacts**
>
> Thank you for pointing this out. We acknowledge that some visual artifacts remain, especially in challenging areas like edges or low-texture regions. However, this task is inherently difficult; the defocused event streams are sparse, noisy, and lack full spatial information, making accurate reconstruction very challenging.
> Despite this, our method achieves significantly better perceptual quality than existing baselines, as shown by multiple no-reference metrics in the tables. We will add more 3D scene results in the revised version.
>
> ---
> ### **Model and Dataset Design**
>
> Thank you for the comments.
> Our synthetic dataset is built on a physically-inspired defocus model that simulates event generation under varying PSF sizes and lighting conditions. By changing blur levels, motion speed, and intensity across sequences, we mimic different scene depths and real-world defocus effects.
> This design allows the model to learn both temporal accumulation and spatial defocus patterns, and to adapt to the link between blur size and event sparsity, making it effective for depth-varying defocus.
>
> We conducted experiments on multi-depth scenes and observed that E2VID pretrained on sharp data can handle slight defocus, suggesting that events exhibit robustness to slight defocus blur. Combined with our specially designed dataset, our model further learns depth-dependent defocus patterns and can distinguish and reconstruct objects at different depths, showing strong generalization to 3D scenarios.
>
> ---
>
> ### **Related work**
>
> Thank you for this suggestion. We will discuss relevant references (e.g., Lou et al., 2023; Lin et al., 2022; Teng et al., 2024) on event-driven autofocus and focal stack fusion. Our method differs in that it does not require active lens control or RGB images, and is designed for passive, single-shot defocused event streams, which we believe complements autofocus-based methods.
>
> ---
>
> Thank you again for your valuable feedback. If there are any additional questions, we would be happy to address them.

---

### Official Review · Reviewer_MV6M · 2025-03-12

**Overall Recommendation:** 4

**Summary:**

This paper introduce a new network architecture for restoring all-in-focus grayscale video from defocused event-camera measurements. The method assumes a thin lens model and gaussian defocus blur. It is tested on simulated and real data. The proposed method outperforms existing methods quantitatively and qualitatively, though not always by a large margin.

Overall, this is a clearly written and technically sound paper that tackles an under-investigated real-world problem.

**Claims And Evidence:**

The method is tested on both simulated and captured event data. The method is tested against two widely used event-to-video restoration networks, which are fine-tuned with simulated defocused data.

Adding an additional baseline where defocused video was restored with a conventional event-to-video architecture and then made all-in-focus with an image-to-image or video-to-video network may improve the paper.

**Essential References Not Discussed:**

Though the problems aren't identical, the following paper is related and should be discussed:
Lou, Hanyue, et al. "All-in-focus imaging from event focal stack." Proceedings of the IEEE/CVF Conference on Computer Vision and Pattern Recognition. 2023.

**Experimental Designs Or Analyses:**

Experiments seemed appropriate.

**Methods And Evaluation Criteria:**

The methods and evaluation seem appropriate.

A one to two sentence description of what the Brisque and Niqe metrics are computing would improve the paper.

**Other Comments Or Suggestions:**

None

**Other Strengths And Weaknesses:**

None noted

**Questions For Authors:**

How do the runtimes of the various methods compare? Can the method be extended to handle events generated by a changing focal plane?

**Relation To Broader Scientific Literature:**

In general, the paper presents an accurate overview of the existing literature. To my knowledge, this exact problem has not been previously addressed.

**Theoretical Claims:**

The derivations seemed correct, though I did not check carefully.

---

> ### Author Rebuttal · Authors · 2025-04-01
>
> **We sincerely thank the reviewer for the encouraging feedback and constructive suggestions. We address the main points below.**
>
> ---
>
> ### **Additional baseline: event-to-video + image/video deblurring**
>
> We appreciate this valuable suggestion. In the supplementary material (Fig. 11–12), we included an experiment that follows a similar two-stage baseline: we first reconstruct intensity images from defocused events, followed by applying NRKNet [CVPR'23]  as the SIDD network for defocus deblurring.
>
> However, the results were suboptimal, mainly due to the low quality of reconstructed images from sparse defocused events, which contain significant artifacts and structural degradation.
>
> These observations support our decision to address defocus directly in the event domain, which our method handles more effectively in terms of both reconstruction fidelity and robustness. We will add more discussion in the revised version.
>
> ---
>
> ### **Clarification on Brisque and NIQE metrics**
>
> Thank you for pointing this out. We will add the following clarifications in the final version:
>
> - BRISQUE (Blind/Referenceless Image Spatial Quality Evaluator) evaluates natural scene statistics in the spatial domain without requiring a reference image.
> - NIQE (Natural Image Quality Evaluator) computes quality based on deviations from learned statistical regularities of natural images.
>
> These no-reference metrics are particularly useful for assessing real-world reconstruction results where ground truth is unavailable.
>
> ---
>
> ### **Discussion of related work: Lou et al. (CVPR 2023)**
>
> Thank you for highlighting this relevant work. Lou et al. [CVPR'23] propose an all-in-focus imaging method based on both RGB images and event focal stacks, which relies on capturing event streams under different focal settings and performing temporal fusion.
>
> In contrast, our work focuses on single-shot defocused event streams, which:
>
> - Require no active focus control or RGB supervision;
> - Address the case where defocus has already occurred, aiming to restore sharp content purely from event data.
>
> We will incorporate a discussion of this difference and clarify the complementary nature of the two approaches in the final revision.
>
> ---
>
> ### **Runtime comparison**
>
> Thanks for your suggestion. We evaluated inference speed and model size under identical settings (input: `[1, 5, 264, 352]`, GPU: RTX 3090). The results are as follows:
>
> | **Model**     | **#Params (M)** | **Inference Time (ms)** |
> |---------------|------------------|--------------------------|
> | E2VID         | 10.71            | 4.41                     |
> | ET-NET        | 22.18            | 25.71                    |
> | SPADE-E2VID   | 11.46            | 12.66                    |
> | **EvFocus**   | **12.11**        | **24.09**                |
>
> Our method offers a balance between performance and complexity. While not the fastest, EvFocus yields consistently higher-quality reconstructions.
>
> ---
>
> ### **Extension to changing focal planes**
>
> Thanks for your advice. This is an excellent and challenging direction. Our current method assumes static defocus, but we believe it can be extended to handle dynamic focal planes by:
>
> - Incorporating temporal modulation or adaptive kernel modeling;
> - Simulating changing focal depth in training data;
> - Leveraging the temporal encoder to track evolving blur distributions.
>
> We consider this a promising line of future work and greatly appreciate the suggestion.
>
> ---
>
> Thank you again for your thoughtful review and support. We are happy to clarify further if any questions remain.

---

### Official Review · Reviewer_hGCk · 2025-03-14

**Overall Recommendation:** 4

**Summary:**

This paper proposes EvFocus, a novel architecture for reconstructing sharp images from defocused event streams. The key innovation lies in its temporal encoder, blur-aware dual-branch decoder, and re-defocus module, combined with a synthetic defocus event dataset for training. Experiments on synthetic and real-world datasets demonstrate superior performance over existing methods under varying blur sizes and lighting conditions.

## update after rebuttal
The author's response has resolved my issues. Considering the other reviewers' comments, I have decided to increase the score.

**Claims And Evidence:**

yes

**Essential References Not Discussed:**

n/a

**Experimental Designs Or Analyses:**

yes

**Methods And Evaluation Criteria:**

yes

**Other Comments Or Suggestions:**

n/a

**Other Strengths And Weaknesses:**

Other Strengths:

First systematic solution for event-based defocus deblurring.

Main Weaknesses:

Computational efficiency and real-time performance metrics are not reported, crucial for event camera applications.

**Questions For Authors:**

1. Why isn't the real defocus ground truth (gt) used during refocusing?

2. What would be the performance if reconstruction is performed first, followed by applying state-of-the-art defocus algorithms in the image domain?

3. What is the significance of using an event camera for this task? Some work [1] suggest that the high-speed characteristics of event cameras enable rapid autofocus to avoid defocus issues. In what scenarios would this algorithm be necessary?

[1] Lou, Hanyue, et al. "All-in-focus imaging from event focal stack." Proceedings of the IEEE/CVF Conference on Computer Vision and Pattern Recognition. 2023.

**Relation To Broader Scientific Literature:**

n/a

**Theoretical Claims:**

yes

---

> ### Author Rebuttal · Authors · 2025-04-01
>
> **We thank the reviewer for their thoughtful comments and constructive suggestions. We address each of the raised points below.**
>
> ---
> ### **Computational Efficiency**
>
> We agree that runtime performance is important for real-world deployment. In response, we have conducted a **comprehensive runtime and parameter comparison** (input size: `[1, 5, 264, 352]` on RTX 3090 GPU):
>
> | **Model**     | **#Params (M)** | **Inference Time (ms)** |
> |---------------|------------------|--------------------------|
> | E2VID         | 10.71            | 4.41                     |
> | ET-NET        | 22.18            | 25.71                    |
> | SPADE-E2VID   | 11.46            | 12.66                    |
> | **EvFocus**   | **12.11**        | **24.09**                |
>
> Our model achieves a good trade-off between accuracy and complexity. The inference time is similar to ET-NET; our method offers significantly improved image quality.
>
> ---
>
> ### **Why isn't the real defocus ground truth (GT) used during refocusing?**
>
> Thank you for this insightful question. Our model architecture includes a blur-aware dual-branch decoder, where the blur decoder passes through three CF modules and fuses with features from the align decoder. We employ a deblurring reconstruction module on the align path to enhance its output. To further leverage the design, we introduce a loss between the blur decoder output and the re-defocus module to encourage CF modules to extract meaningful blur representations.
>
> One of our intentions is to encourage effective feature interaction between the two branches. Specifically, the blur decoder extracts blur-aware event representations, which are used to guide and support the learning process of the align decoder. This interaction allows the align decoder to better capture defocus-related features from the event domain.
> In fact, there is a modality gap between events and RGB images: some low-texture regions or subtle motion details present in the GT image may not be well represented in event streams. In this context, directly supervising with defocus GT could potentially lead the optimization to overly favor one branch (e.g., the blur decoder), focusing more on image-domain appearance rather than learning the common defocus patterns reflected in the event stream. This may weaken the collaborative learning dynamics between the two branches.
>
> Our current design aims to balance the contributions of both branches and promote cooperative learning, and has proven effective in practice (validated through ablation studies in the paper).
> We appreciate the reviewer’s question, and we will include a more detailed explanation of this design choice in the revised version.
>
> ---
>
> ### **What would be the performance if reconstruction is followed by defocus deblurring in the image domain?**
>
> We performed experiments and reported results in our supplementary material (Fig. 11 and Fig. 12), where the NRKNet network was employed as SIDD network processing for defocus removal.
>
> The results of other methods + SIDD were suboptimal. This is because:
>
> - Defocused event streams are inherently sparse and contain degraded spatial gradients.
> - Reconstructed intensity images from such events exhibit heavy artifacts and blurred textures, limiting the effectiveness of downstream image-based deblurring algorithms.
> - Events capture log-intensity changes, not absolute intensity, and are affected by defocus noise, which severely impacts the quality of intermediate reconstructions.
>
> These findings suggest that directly addressing defocus in the event domain, as EvFocus does, is more robust and efficient.
>
> ---
>
> ### **What is the significance of using an event camera for this task?**
>
> Thank you for your suggestion. Compared to Lou et al. [1], their work leverages a focal sweep with event cameras to reconstruct an all-in-focus image using timestamp selection and focal stack merging. This method assumes a dynamic focus adjustment and high-quality RGB input.
> In contrast, our setting targets scenarios where only event streams are available, especially in high-speed or low-light scenes.
>
> Our approach reconstructs sharp images from single-pass, defocused event streams, making it suitable for settings without active focus control or RGB inputs like Lou et al. [1]. Hence, it complements Lou et al.’s method and serves different application domains.
>
> ---
>
> We thank the reviewer again for the helpful comments and the recognition of our work. We will incorporate the suggestions and clarifications in the revised version.

---

### Decision · Program_Chairs · 2025-05-01

**Decision:**

Accept (poster)

**Comment:**

This paper received three positive initial reviews. There was general consensus about the importance of the problem considered (video reconstruction from event streams under motion blur), the technical soundness of the proposed solution, and the clarity of presentation. There were some concerns raised about experimental evaluations (e.g., baseline comparisons, choice of metrics, and overall visual quality of the reconstructed results).

The authors provided a comprehensive response, after which two of the reviewers maintained positive (accept) decisions.

One of the reviewers, however, lowered their score citing low computational efficiency of the proposed method (and some other technical concerns). These concerns, while valid, are not considered prohibitive.

Therefore, on balance, an accept decision was reached.